# Neuronavigated Cerebellar 50 Hz tACS: Attenuation of Stimulation Effects by Motor Sequence Learning

**DOI:** 10.3390/biomedicines11082218

**Published:** 2023-08-08

**Authors:** Rebecca Herzog, Christina Bolte, Jan-Ole Radecke, Kathinka von Möller, Rebekka Lencer, Elinor Tzvi, Alexander Münchau, Tobias Bäumer, Anne Weissbach

**Affiliations:** 1Institute of Systems Motor Science, University of Lübeck, Ratzeburger Allee 160, 23562 Lübeck, Germany; rebecca.herzog@uni-luebeck.de (R.H.); c.bolte@uni-luebeck.de (C.B.);; 2Center of Brain, Behavior and Metabolism (CBBM), University of Lübeck, Ratzeburger Allee 160, 23562 Lübeck, Germany; 3Department of Neurology, University Hospital Schleswig Holstein, Ratzeburger Allee 160, 23562 Lübeck, Germany; 4Department of Psychiatry and Psychotherapy, University of Lübeck, Ratzeburger Allee 160, 23562 Lübeck, Germany; 5Department of Neurology, Leipzig University, Liebigstraße 20, 04103 Leipzig, Germany; 6Syte Institute, Hohe Bleichen 8, 20354 Hamburg, Germany

**Keywords:** cerebellum, transcranial alternating current stimulation, tACS, motor sequence learning, MSL, neuronavigation, electric field simulation

## Abstract

Cerebellar transcranial alternating current stimulation (tACS) is an emerging non-invasive technique that induces electric fields to modulate cerebellar function. Although the effect of cortical tACS seems to be state-dependent, the impact of concurrent motor activation and the duration of stimulation on the effects of cerebellar tACS has not yet been examined. In our study, 20 healthy subjects received neuronavigated 50 Hz cerebellar tACS for 40 s or 20 min, each during performance using a motor sequence learning task (MSL) and at rest. We measured the motor evoked potential (MEP) before and at two time points after tACS application to assess corticospinal excitability. Additionally, we investigated the online effect of tACS on MSL. Individual electric field simulations were computed to evaluate the distribution of electric fields, showing a focal electric field in the right cerebellar hemisphere with the highest intensities in lobe VIIb, VIII and IX. Corticospinal excitability was only increased after tACS was applied for 40 s or 20 min at rest, and motor activation during tACS (MSL) cancelled this effect. In addition, performance was better (shorter reaction times) for the learned sequences after 20 min of tACS, indicating more pronounced learning under 20 min of tACS compared to tACS applied only in the first 40 s.

## 1. Introduction

Transcranial alternating current stimulation (tACS) is a widely used method of non-invasive brain stimulation that involves the application of a low-intensity current alternating in a defined frequency between at least two electrodes on the subject’s scalp. The position of the electrodes, as well as the frequency and amplitude of the alternating current, depends on the study objective, target region or cell type. The exact mechanisms by which tACS affects brain functioning and oscillations are not yet fully understood. However, a local effect on the spike-timing of neuron populations in the target area of the electric field is assumed [1,2,3], influencing the magnitude of brain oscillations [4]. Furthermore, there are also indications of network effects that can extend beyond the electric field in the targeted brain area [5].

tACS effects, in general, can occur “online” during stimulation and “offline” following stimulation, indicating persisting effects. “Offline” effects are likely due to spike-timing-dependent plasticity, which may link immediate entrainment during stimulation with longer-lasting long-term potentiation (LTP) and long-term depression (LTD) mechanisms [6,7,8,9]. Importantly, previous findings suggest that tACS effects depend on the state of the brain, i.e., the interaction between endogenous neural activity in the targeted brain area and exogenous electric stimulation [10,11,12]. The possibility of targeting different neuronal populations or networks as well as processes that are linked to specific oscillation frequencies could represent an advantage over other methods of electrical stimulation. Therefore, a precise understanding of the underlying mechanisms is indispensable [13]. Neocortical gamma oscillations (~25–100 Hz) are thought to play a key role in movement and have been associated with a pro-kinetic effects [14,15] that could be modulated by gamma tACS, suggesting a causal link between gamma band oscillations and the facilitation of motor processing in different tasks [16,17,18]. However, these findings are largely limited to the M1.

The cerebellum is an important relay for a broad range of brain functions, such as motor control, sensorimotor integration, language and emotional processing [19]. Abnormal cerebellar activity has been documented in a number of movement disorders, including tremors, dystonia and myoclonus [20,21,22], rendering it an interesting target for neuromodulation. Especially, the possibility of the frequency-specific targeting of a region of interest using tACS could be an advantage compared to other neuromodulation methods.

Cerebellar tACS in various frequencies ranging from 5 to 300 Hz has been examined in different protocols and studies with heterogenous results [23]. Fifty Hz tACS has been suggested to correspond to the endogenous frequency of the Purkinje cells [24]. In a previous study from our lab, cerebellar tACS influenced the motor system most strongly and persistently compared to other forms of cerebellar transcranial current stimulation, causing increased excitability of the primary motor cortex (M1) for at least one hour after stimulation [25]. Naro et al. reported an increase in motor cortex excitability and a reduction in cerebellar brain inhibition, whereas Spampinato et al. did not find any changes after cerebellar 50 Hz tACS [26,27]. On a behavioral level, cerebellar 50 Hz tACS improved the motor function of the upper limbs but not explosive power (the ability to produce a maximum amount of force in a very short period of time) in sports performance [26,28]. To our knowledge, the influence of the simultaneous activation of the motor system on the effects of 50 Hz cerebellar tACS is unclear. The heterogeneity of the results could be explained by specific challenges of cerebellar tACS, in particular by the complex anatomy of the cerebellum. First, the cerebellum has a different cytoarchitecture than that of the neocortex, which limits the generalization of the findings from cortical tACS to cerebellar tACS and vice versa [29]. Second, the cerebellum has a complex surface structure and a high degree of functional specializations within an overall small volume [30]. Thus, the target accuracy/focality of transcranial electrical stimulation, including the electrode position, as well as the individual anatomy, plays an important role [31,32]. 

Electrode placement can be chosen based on anatomical landmarks on the skull or individually based on MRI neuronavigation. Previous electric field simulations with anatomically positioned electrodes showed that the Celnik montage, where one electrode is placed 3 cm lateral to the inion and the other electrode is placed on the ipsilateral buccinator muscle, affected the lobules Crus I/II, VIIb, VIII and IX of the cerebellar hemisphere without significant spread to the contralateral hemisphere [33]. Neuronavigated electrode placement may allow more targeted stimulation. However, this has not been investigated yet, and it has not it been simulated or tested experimentally either. 

In this study, we (1) investigated the effects of cerebellar tACS on corticospinal excitability and motor sequence learning (MSL), (2) determined whether MSL during cerebellar tACS alters the effect of tACS and (3) evaluated neuronavigated electrode montages via the post hoc simulation of individual electric fields based on T1-weighted MRI data in a group of healthy subjects. As a measure of corticospinal excitability, we determined the motor evoked potential (MEP) evoked by single-pulse transcranial magnetic stimulation (TMS) over the left primary motor cortex (M1) at two different time points after tACS. Reaction times in MSL were determined to assess tACS online effects, because MSL has been linked to activation in multiple cortical and subcortical brain regions, including the cerebellum [34,35,36]. MSL has been shown to be affected by cerebellar tACS [4,37].

## 2. Materials and Methods

### 2.1. Study Design and Participants

We recruited 25 healthy, right-handed subjects without self-reported neurologic or psychiatric disorders. Subjects with epilepsy or structural brain injury, pregnant people and participants who either reported frequent gaming or playing an instrument regularly were not included. Due to dropout during the experiments, 20 participants (13 female, 7 male, mean age of 26.5 years) were included in the analysis. 

All subjects were investigated in four sessions on separate days to evaluate 50 Hz tACS over the right cerebellar hemisphere for 40 s (originally intended to be sham; short (s)-tACS) or 20 min (long (l)-tACS). Stimulation was applied either at rest (l-tACS_Rest_, s-tACS_Rest_) or during the MSL task (l-tACS_MSL_, s-tACS_MSL_, see Figure 1). The interval between interventions was at least one week to avoid carry-over effects. The order of intervention was balanced between subjects. 

In every session, MEP amplitudes were determined at three time points: one before and two after tACS (post 1: approx. 5 min; post 2: approx. 50 min after stimulation). Reaction times during MSL were used as a readout for the online effects of tACS. After each session, participants completed a questionnaire addressing the side effects of the neuromodulation (vertigo, change in coordination ability) of tACS application (local heat, skin sensations) and of TMS (headache). Potential side effects were rated on a visual analog scale from 0 to 10, with 0 indicating “no effect” and 10 indicating “the strongest imaginable effect”. 

### 2.2. Neuronavigation and MRI Data Acquisition

To target the stimulation site precisely, we used the Brainsight neuronavigation system (Rogue Research, Montreal, QC, Canada) in combination with the Polaris camera (Northern Digital, ON, Canada) as in our previous studies [25,38]. The stimulation sites were previously marked in an individual T1-weighted MRI scan. The neuronavigated localization of the left M1 was verified by determining the neurophysiological “motor hot spot”, i.e., the location where TMS pulses administered at a supra-threshold intensity consistently produced the largest MEPs. 

For targeted cerebellar tACS, we chose lobule VIIIA because it has been shown to be important for the execution of motor tasks and learning processes [19,39]. Due to its superficial location, it is reachable and has already been used as a target in other TMS and tACS studies [25,40]. 

Structural MRI data were acquired at the Center of Brain Behavior and Metabolism (CBBM) Core Facility Magnetic Resonance Imaging using a 3-T Siemens Magnetom Skyra scanner (Siemens, Erlangen, Germany) equipped with a 64-channel head-coil. Structural images of the whole brain using a 3D T1-weighted MP-RAGE sequence were acquired (TR = 1900 ms; TE = 2.44 ms; TI = 900 ms; flip angle 9°; 1 × 1 × 1 mm^3^ resolution; 192 × 256 × 256 mm^3^ field of view; acquisition time of 4.5 min).

### 2.3. Transcranial Magnetic Stimulation Measurements

The experimental setup for TMS was similar to that in previously published TMS studies [25,41,42]. Each subject was seated in a comfortable position. For relaxed sitting, the arms were placed on a cushion if necessary. Participants were instructed to sit relaxed and keep their eyes open. Electromyography (EMG) of the right first dorsal interosseus muscles was captured using Ag/Ag-Cl disc surface electrodes in a belly tendon montage, and the ground electrode was fixed at the wrist. A D360 amplifier (Digitimer Limited, Welwyn Garden City, Hertfordshire, UK) was used to filter (High pass: 10 Hz, low pass: 1 kHz, 50 Hz Notch) and amplify EMG signals. With a laboratory interface (Micro 1401; Cambridge Electronics Design, Cambridge, UK) the EMG signal was digitized at a 5 kHz sampling rate and recorded. Data were stored on a personal computer using the SIGNAL software (Version 6.06, Cambridge Electronic Devices, Cambridge, UK).

TMS pulses were generated by a Magstim 200^2^ stimulator (Magstim Company, Whitland, Dyfed, UK). The left M1 was stimulated with a 70 mm figure-eight-shaped coil (Magstim Company, Whitland, Dyfed, UK). 

MEPs were generated by single-pulse TMS at a supra-threshold intensity, evoking a mean MEP of about 1 mV. Before and after the intervention, the same TMS intensity was used for the measurements of unconditioned MEPs. At each time point (pre, post 1, post 2), 15 MEPs were collected. Motor thresholds were recorded as part of a second experiment. Here, the resting motor threshold (RMT) was recorded with a 70 mm coil with an anterior-posterior current direction (only at two time points: pre and post 1), and the active motor threshold (AMT) was recorded with a 25 mm coil with a posterior-anterior current direction.

### 2.4. Transcranial Alternating Current Stimulation 

tACS was applied with a DC-Stimulator plus (neuroCare, Munich, Germany) using the “study mode” to guarantee a double-blind design. We decided to stimulate for 20 min because this has been shown to be safe and effective in earlier studies [25,38]. Sham stimulation was defined as stimulation with the same parameters as those of the real stimulation, apart from the duration (40 s, i.e., 1/30 of the verum stimulation). Unexpectedly, we also found plasticity effects in this 40 s condition (see results), so this condition cannot reasonably be considered a sham condition. Therefore, to avoid confusion, we do not refer to this condition as a sham condition in the following. 

In each session, we used the same neuronavigated electrode montage. Using the Brainsight neuronavigation system, one electrode was placed over the right cerebellum with the center of the electrode at the radial scalp projection of the individually neuronavigated location of the target lobule VIIIA. The second electrode was placed on the right cheek (“Celnik-Montage”). An alternating current with a peak-to-peak intensity of 1 mA was applied via 3 × 3 cm rubber electrodes evenly covered with conductive paste (Ten20 conductive electrode paste; Weaver and Company, Aurora, CO, USA, current density: 0.11 mA/cm^2^). We used conductive paste instead of saline-soaked sponges because we feared that the sponges could dry out due to the relatively long duration of the stimulation. The frequency was set at 50 Hz. Fade in/out was set to 100 × 2π cycles (2 s). During stimulation, the subjects were either instructed to sit on a chair without moving or talking or to perform the MSL task during the stimulation. 

### 2.5. Motor Sequence Learning

For MSL, we used a variant of the serial reaction time task (SRTT, [43]). In this task, subjects are asked to respond to a visual cue that changes from trial to trial by pressing a button. For the presentation of the stimuli, we used the software “Presentation” (Version 22.0, Neurobehavioral Systems, Albany, NY, USA). In every trial, four black squares were presented in a horizontal order, with every square corresponding to one key. The corresponding keys on the keyboard were the numbers 7, 8, 9 and 0. The subjects were instructed to position their fingers on the marked keys on a laptop keyboard (Lenovo ThinkPad L540, Lenovo Group Limited, Hongkong, China). At stimulus onset, one of the squares turned blue to indicate that the subject should press the corresponding key as fast as possible (see Figure 2). If the subject pressed the wrong key, the square turned red, and if the subject was too slow (i.e., reaction time > 1000 ms), the words “too slow” were shown (in German). 

The SRTT had three conditions: Simple (SMP), Random (RND) and Sequence (SEQ). Every session began with 80 SMP trials in which the keys had to be pressed in descending order (4-3-2-1-4-3-2-1). After that, RND and SEQ blocks alternated. In RND blocks, the stimuli were pseudorandomized (80 trials), and in SEQ blocks, a sequence of eight items was repeated 15 times (120 trials). RND and SEQ blocks were designed to avoid any association effect (balanced frequency of items and pairs; see [44]). Overall, the task consisted of 11 blocks, 1 SMP block, 5 RND blocks and 5 SEQ blocks. After each block, the subject had a break of 20 s. The task lasted 20 min, e.g., as long as the stimulation with l-tACS, and was performed simultaneously. 

To avoid any cross-over learning effects between the sessions, we used two different sequences for the two MSL sessions. The sequences were balanced across stimulation protocols.

### 2.6. Data Analysis and Statistical Analysis

#### 2.6.1. Motor Evoked Potentials

MEP peak-to-peak amplitudes were measured in each trial. For statistical analysis, a multifactorial analysis of variance with repeated measures (ANOVA) using the factors of Intervention (l-tACS, s-tACS), MSL (MSL/no MSL during tACS application) and Time (pre, post 1, post 2) was performed. If the ANOVA indicated a significant *p*-value with *p* ≤ 0.05 for a main effect or interaction, a Bonferroni–Holm-corrected paired Student’s *t*-test was used for post hoc testing. We performed Spearman–Rho correlation analyses for all neurophysiological and behavioral parameters as well as electric field magnitudes.

#### 2.6.2. Motor Sequence Learning

Data analysis was performed in MATLAB (version R2021b; The Mathworks, Natick, MA, USA) and R (version 2022.07.1; R Foundation for Statistical Computing, Vienna, Austria). Median response times (RT) of the MSL task were determined as an outcome parameter during tACS. To calculate RTs, 40 trials were first combined into a mini block to evaluate changes over time and avoid learning and fatigue effects. This resulted in two mini-blocks per RND block and three mini-blocks per SEQ block (corresponding to five repetitions of the corresponding sequence). SMP was not included in the analysis. First, the median RT was calculated across the 40 trials, excluding both runs and including errors and runs with RTs exceeding 2.7 standard deviations (outermost ~ 1% of the normal distribution) of the median RT of the mini block. 

For statistical analysis, a linear mixed model was used. As fixed effects, Intervention (s-tACS, l-tACS) and Block (1–10, with Block 1,3,5,7,8 being RND blocks and 2,4,6,8,10 being SEQ blocks) were included. Subjects were included as random effects. An ANOVA of the fixed effects was performed.

#### 2.6.3. Evaluation of Side Effects

At the end of each measurement day, the participants were asked to fill out a questionnaire regarding the side effects that they had experienced. They were asked to rate possible side effects (headache, discomfort, dizziness, local heating) on a scale of 0–10 (0 = not at all, 10 = very strong). To analyze the distribution of side effects as well as the most/least comfortable stimulation, we performed Χ^2^ goodness of fit tests for each measurement day (l-tACS_MSL_, l-tACS_Rest_, s-tACS_MSL_, s-tACS_Rest_), hypothesizing a balanced frequency. 

#### 2.6.4. Electric Field Simulations

Individual electric field simulations were computed based on T1-weighted MRI data using the ROAST toolbox for MATLAB (The Mathworks Ltd., Novi, MI, USA) [45,46]. First, T1 images were normalized to the ICBM152 standard brain using SPM12 (www.fil.ion.ucl.ac.uk/spm, accessed on 14 August 2022). Central points of the stimulation electrodes on the scalp were registered using the Brainsight neuronavigation system (Rogue Research, Montreal, QC, Canada). Positions were projected to the scalp surface and converted to MNI voxel space to align with the respective normalized T1 image. Subsequently, individual T1 images and electrode positions were submitted to ROAST to segment the MRI into six compartments, assigning tissue-wise conductivities for skin (0.465 S/m), bone (0.01 S/m), air (2.5 × 10^−14^ S/m), cerebrospinal fluid (CSF, 1.65 S/m), white matter (0.126 S/m) and gray matter (0.276 S/m). Electrodes were simulated in a complete electrode model as 3 × 3 cm patches centered around the neuronavigated electrode positions on the scalp surface, oriented along the left-right axis for electrodes placed over the right cerebellum (1 mA) and with an anterior-posterior orientation for electrodes placed over the right m. buccinator (−1 mA; gel: 0.3 S/m, electrode: 5.9 × 10^7^ S/m. Tetrahedral meshes and forward modeling of electric field simulations using the finite element method (FEM) were computed using the default parameters implemented in ROAST. As an output, ROAST provides the electric vector field interpolated to voxel space.

The whole-brain electric field magnitude |E| was quantified as the vector length at each voxel $\vec{x}$ of the electric field $\overset{\text{→}}{E}$ as $\sqrt{{\left[\vec{E}\left(\vec{x}\right)\right]}^{2}}$. For the analysis of electric field intensities, tissue-wise electric field magnitudes (|E|_kmax_), as well as the target intensity (|E|_target_), spatial extent (|E|_extent_) and targeting bias (|E|_bias_), were computed for each region of interest (ROI) defined by the Automated Anatomical Labeling (AAL) atlas [47]. Whereas target intensity and spatial extent reflect the magnitude in each ROI, spatial extent and targeting bias indicate the focality and spatial specificity of the electric field, relative to each ROI (see Appendix A for details). 

Target intensity (|E|_target_) and bias (|E|_bias_) for nine putative stimulation targets in the lateralized cerebellar ROIs were subjected to separate repeated-measures ANOVAs, including the factors of ROI (cerebellar ROIs: Crus I, Crus II, III, IV/V, VI, VIIb, VIII, IX, X) and hemisphere (left, right). Greenhouse–Geisser correction was applied if the sphericity assumption was violated. Post hoc *t*-tests were computed (two-sided), and the effect sizes as well as Bonferroni-corrected *p*-values are reported in the case of significant main or interaction effects. Significance levels were set to α = 0.05. 

For illustration, electric field magnitudes in gray matter were interpolated on the cortical surface of the MNI ICBM152 standard brain, as well as on the cerebellar surface from the SUIT toolbox using a spatial Gaussian filter (5 mm width) [48]. Furthermore, gray matter electric field magnitudes were interpolated on a template cerebellar flatmap representation [48]. 

## 3. Results

### 3.1. Cortical Excitability Measures

The multifactorial ANOVA of MEP amplitudes showed no three-way interaction of the factors of MSL, Time and Intervention (F_2,38_ = 0.221, *p* = 0.803, η^2^_p_ = 0.011), suggesting no differential effect of the interventions. Both interactions with Intervention as a factor (MSL x Intervention, Time x Intervention) were not significant (*p* = 0.606, *p* = 0.322, respectively). However, a significant interaction was found for MSL x Time (F_2,38_ = 4.58, *p* = 0.017, η^2^_p_ = 0.194), indicating that the effect of tACS—regardless of the duration of the stimulation—was influenced by the presence or absence of MSL. Post hoc significant differences between no MSL pre and no MSL post 1 (t_19_ = −3.59, *p* = 0.04) and between MSL pre and no MSL post 1 (t_19_ = 3.46, *p* = 0.037) were found. Stimulator output to achieve 1 mV MEP did not differ significantly at baseline (F_3,57_ = 0.48, *p* = 0.697). There was no effect of Intervention on motor thresholds, with the ANOVAs revealing no interaction between Time and Method for either RMT or AMT (F_6,114_ = 1.67, *p* = 0.136; F_3,45_ = 0.21, *p* = 0.891, respectively). For descriptive data, see Appendix A.

Taken together, we saw an increase in motor cortex excitability after tACS, but only if the subject did not perform MSL during the stimulation. On a descriptive level, the increase persisted longer after l-tACS than after s-tACS (see Figure 3). 

### 3.2. Motor Sequence Learning

The ANOVA of the fixed effects revealed a significant interaction of Intervention and Block (F_9,962_ = 2.57, *p* = 0.009), with faster reaction times during l-tACS in the last SEQ block (t_961_ = 4.21, *p* = 0.005, see Figure 4).

### 3.3. Electric Field Simulation

Electric field simulations of this study provide systematic evidence for the spatial specificity of electric fields induced by tACS targeting the right cerebellar region VIII (Figure 5). Neuronavigation suggested the placement of electrodes over the right cerebellum with the center point at [14, −115, −37] ± [4, 2, 6] (MNI-coordinates). Assuming that the inion was at [−1, −121, −20] [49], in the present study, the average electrode center was placed 1.6 cm to the right and 1.5 cm below, relative to the inion (2.3 ± 0.6 cm Euclidean distance).

Electric field magnitudes in gray matter were estimated at 0.11 ± 0.01 V/m (|E|_kmax_, M ± SD). Overall electric fields were well targeted to the right cerebellum (Figure 5), ranging from 0.07 to 0.13 V/m in the cerebellum (Appendix A), including the vermis (Appendix A). The highest electric field magnitudes were observed in areas VIIb (0.12 ± 0.02), VIII (0.12 ± 0.01) and IX (0.13 ± 0.01) in the right cerebellar hemisphere. Inter-individual variability ranged from 0.09 to 0.13 V/m, deviating up to 18% from the average electric field magnitude (see Appendix A). Descriptive spatial extent (35 mm averaged across all right cerebellar ROIs) indicated quite focal electric fields (compare [50]). Targeting bias was the smallest for areas VIII (18 ± 6 mm) and IX (12 ± 4.1 mm) of the right cerebellar hemisphere, indicating the targeting of electric fields to these areas. Detailed results are provided in the Appendix A. 

To estimate the targeting specificity across cerebellar ROIs, |E|_bias_ was computed, quantifying the focus of the electric field with respect to the median coordinate of each cerebellar ROI. The repeated-measures ANOVA revealed a significant ROI x Hemisphere interaction (F_3.2,60.3_ = 8.18, *p* < 0.001, $\eta_{p}^{2}$ = 0.301) and significant main effects of both ROI and Hemisphere (*p* < 0.001). Post hoc *t*-tests indicated lower biases for the right hemisphere compared with the left hemisphere across all lateralized cerebellar ROIs (all *p* < 0.018, all d > 0.81), except for area III (*p* = 0.828). Bias was the lowest for right area IX compared to all other ROIs. Right area XIII showed lower bias compared to all other regions, except right Crus II and area X (Figure 5E). For further details, see Appendix A.

As indicated by the ANOVA analysis (see Appendix A), the magnitudes of individual electric fields were larger in the right cerebellar hemisphere compared to the left hemisphere with highest intensities in areas IX, VIII and VIIb (Figure 5, see Appendix A). In addition, analysis of the electric field bias supports the conclusion that the electric field was mainly targeted to areas IX and VIII of the right cerebellar hemisphere (Figure 5E, see Appendix A).

### 3.4. Correlation Analyses

The Spearman–Rho correlations revealed no relevant significant correlations (data are available upon request).

### 3.5. Evaluation of Side Effects

The side effects of electrostimulation are listed in Table 1. All answers except 0 were initially evaluated as the occurrence of the side effect. The X^2^ tests estimating the assessed side effects of the stimulation revealed no significant differences between the four sessions. 

Mean values for the assessment of side effects were almost always below 1 across all subjects, i.e., the side effects occurred to a minimal extent overall. One exception was a rating of 7 for “local heating” during stimulation with l-tACS_MSL_, resulting in a mean value of 1.3 for the side effect of this stimulation. No other ratings above 3 occurred. One subject also reported the occurrence of phosphenes during stimulation in all four sessions.

## 4. Discussion

In this study, we aimed for a more in-depth understanding of the mechanisms of action underlying the effects of cerebellar 50 Hz tACS. 

We verified our electrode montage using a simulation of the electric fields, showing that the neuronavigation-based positioning of the electrodes ensured a focal stimulation of the right cerebellum and the targeted cerebellar lobule VIII. Although cerebellar 50 Hz tACS applied at rest increased motor cortex excitability, this effect did not occur in the MSL condition, suggesting an attenuating effect of motor activation on the stimulation efficacy. To our knowledge, this is the first study to examine the modulatory effect of concurrent motor activation (vs. rest) on cerebellar tACS. In addition, tACS slightly improved motor learning performance during the stimulation. Our findings of increased corticospinal excitability after tACS applied at motor rest are in line with those of our own previous study and the study of Naro et al. [25,26]. Moreover, the effect was also evident after only 40 s of tACS.

The increase in the unconditioned MEP may be due to the reduced inhibitory output of the cerebellum, causing disinhibition in cerebello-thalamo-cortical pathways. As tACS interacts with local oscillations, there are multiple possibilities for the increase in MEPs [25]. First, even though one would expect an increase in the activity of Purkinje cells due to resonance effects, the latter can lead to transsynaptic LTD due to network modulation and subsequent biochemical changes [51]. Second, the cells regulating the Purkinje cells may have been influenced by the stimulation as well. Granule cells are molecular layer interneurons with an excitatory output to the Purkinje cells with a natural frequency in the theta band (5–7 Hz) [52]. If their activating output on the Purkinje cells over the parallel fibers is negatively influenced by tACS, it can also lead to a decrease in the inhibitory output of the Purkinje cells and thereby to an increase in corticospinal excitability. 

MSL is defined as the acquisition and optimization of a novel series of movements with practice [53]. The cerebellum plays an important role in predictive movement control and the formation of internal models and is involved in motor learning [54,55]. Internal models map the discrepancies between planned and executed movements and are thought to be encoded by synapses between Purkinje cells and parallel fibers, and in turn, the parallel fiber input is modulated by climbing fibers [56]. Here, LTD is induced when these are activated simultaneously with the climbing fibers. Therefore, the modulation of Purkinje cells could lead to an induction of LTD, which might in turn influence the formation of internal models and thus induce changes in motor learning. 

In the literature, there are few reports regarding 50 Hz tACS of the cerebellum in relation to MSL. Giustiniani et al. found deterioration of reaction times in a serial reaction time task during 50 Hz tACS and argued that this frequency interferes with the formation of internal models in the cerebellum and thus hampers motor learning [37]. Another study found no change in the acquisition or retention of new motor skills during 50 Hz tACS of the cerebellum [57]. In contrast, in our study, improved reaction times for sequences during long 50 Hz tACS were found toward the end of the task.

Possible reasons for the discrepancies between the results may lie in the different methodologies of the studies. Giustiniani et al. applied stimulation for 5 minutes. Wessel et al., on the other hand, applied stimulation for 20 min, which is the same duration as that in the present study, but they used a different motor learning task, which makes this comparison difficult as well. In addition, there are differences in the electrode montage, stimulated cerebellar hemisphere or hand used for the motor task, as well as in the awareness of learning. Taken together, the results provide evidence for the online effects of 50 Hz tACS on the cerebellum and for the functional relevance of this frequency for MSL.

The interaction of brain state and the effect of tACS was described previously [10]. For instance, a previous study examining the effect of 50 Hz cerebellar tACS applied during MSL did not find an effect on corticospinal excitability [37], which is in line with our findings. Another study could show that simultaneous tACS in the γ-range inverts the effect of continuous theta-burst stimulation, which usually induces LTD-like effects, as well [58,59]. These effects could be mediated by homeostatic plasticity, which suppresses plasticity during high-level general network activity by raising the threshold for a subsequent plasticity induction [60]. Similar effects have also been observed when applying stimulation and motor activity sequentially [61].

On a cellular level, the direction and efficacy of a stimulation protocol depends on the excitability of the postsynaptic neuron when the external stimulus is applied, an effect known as “gating” [62]. The excitability of the postsynaptic neuron is mediated by the concentration of intracellular Ca^2+^, which is controlled by the input of intracortical circuits and plays an important role in LTD and LTP [63]. Small changes in intracellular Ca^2+^ concentrations lead to LTD, whereas larger changes in Ca^2+^ concentrations lead to LTP [64]. In our case, both MSL and 50 Hz tACS might have led to an LTD-like effect mediated by a slightly increased intracellular Ca^2+^ level when applied isolated. When applied together, this could have pushed the Ca^2+^ influx in the neurons to a level where no LTD was induced [64,65]. 

An important aspect needs to be kept in mind for the design of subsequent study protocols. Cerebellar tACS applied for 40 sec showed an effect on corticospinal excitability. Other groups used sham stimulation protocols with a stimulation duration of 10 s and 30 s without an effect on corticospinal excitability [26,57,66,67], whereas increased MEP amplitudes after 60 s of stimulation have been reported [66]. Given these findings, there seems to be a narrow time window for sham stimulation. 

To analyze the electric field contribution, we computed individual FEM simulations of electric fields in all twenty subjects. Complementing a previous study of three-compartment BEM electric field simulations in one standard brain [32], our data illustrate the important contribution of including additional tissues in the head model when estimating electric fields in the cerebellum. Especially, the CSF compartment has been shown to affect simulated electric field properties in the cortex [68,69] and the cerebellum [31]. In this study, higher descriptive intensities were observed in deeper cerebellar regions (including the vermis) when compared to previously reported three-compartment BEM simulations [32]. These results indicate that the CSF distribution might allow electric fields to reach areas other than superficial cerebellar regions, especially the caudal cerebellum and including the vermis (Figure 5B,C).

Importantly, individual field simulations indicated areas VIII, IX (and VIIb) in the right cerebellar hemisphere as the main targets of the electric fields induced by the applied stimulation montage neuronavigated to area VIII. Thus, although CSF does have an impact on the current flow, hemisphere-specific targeting of the relatively small cerebellar volume appears feasible (Figure 5D). In addition to the robust estimate of the average electric field (including individual electric field simulations of all subjects), our data also illustrate the high inter-individual variability of electric field intensities in the cerebellum, varying up to 18% from the mean intensity in the gray matter compartment (|E|_kmax_) and similarly from that in cerebellar ROIs (|E|_target_, Figure 5D). A previous study estimated electric variability to be 55% [31]. Because no direct comparison between neuronavigated and conventional electrode placement was assessed in this study, we can only speculate that the descriptively decreased variability was due to the neuronavigated approach that we used. Specifically, the electrode placement might benefit from the neuronavigated approach, in which an individual intracranial target region is defined and cerebellar anatomical details are considered, compared to the electrode placement based on more superficial anatomical landmarks defined on the scalp surface.

Several limitations need to be considered. Because our 40 s condition was revealed to have effects on corticospinal excitability, we treated it as a second intervention. Therefore, strictly speaking, there was no sham condition, and “pure” effects of MSL alone could not be determined with certainty. Moreover, given their intrinsic variability, it would have been preferable to measure a higher number of MEPs. Regarding motor thresholds, we did not use the standard procedure due to study design reasons, resulting in limited interpretability. Moreover, the SRTT task that we used places only minimal demand on motor execution. Therefore, it remains to be determined if and to what extent tACS would affect more complex, real-world tasks. Finally, we only examined the effects of 50 Hz tACS but not those of other frequencies, so the frequency specificity of the observed effects cannot be claimed. Due to the complex design of this study and the risk of higher dropout rates, we could not address this relevant question. This should be addressed in future studies.

## 5. Conclusions

In summary, we found significant effects of cerebellar 50 Hz tACS on corticospinal excitability both after short (40 s) and prolonged (20 min) stimulation. These effects were attenuated by concurrent motor activity. Prolonged but not short 50 Hz tACS improved MSL. In addition, neuronavigated electrode placement enables focal transcranial current stimulation despite the anatomical complexity of the cerebellum.

## Figures and Tables

**Figure 1 biomedicines-11-02218-f001:**
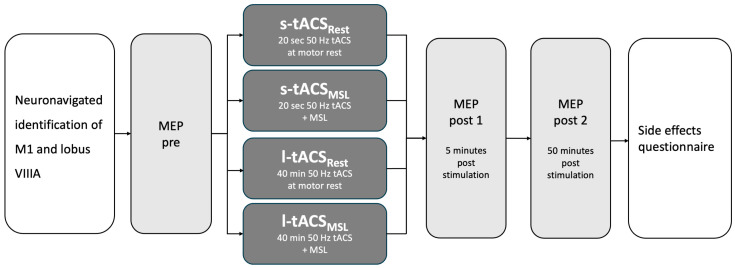
Study design. M1 = primary motor cortex, MEP = motor evoked potential, s-tACS_Rest_ = tACS applied for 40 s at motor rest, s-tACS_MSL_ = tACS applied for 40 s during motor activation, l-tACS_Rest_ = tACS applied for 20 min at motor rest, l-tACS_MSL_ = tACS applied for 20 min during motor activation.

**Figure 2 biomedicines-11-02218-f002:**
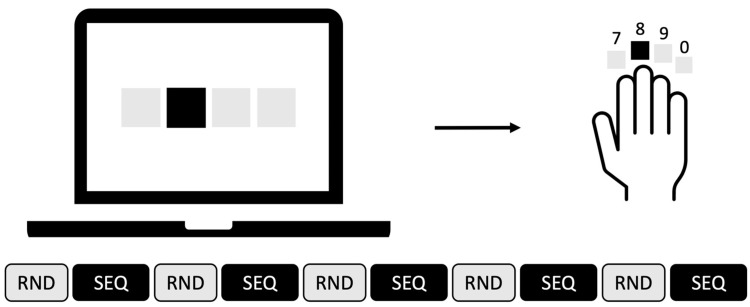
Stimulus material (**top**) of the MSL task during tACS. Sequence of conditions during the task (**bottom**). RND = randomly presented stimuli, SEQ = sequentially presented stimuli.

**Figure 3 biomedicines-11-02218-f003:**
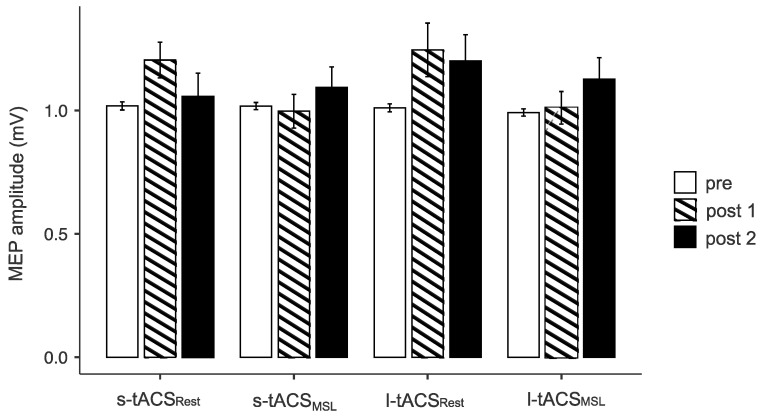
Motor evoked potential (MEP) before and after intervention (post 1 = approx. 5 min and post 2 = approx. 50 min after the end of stimulation). s-tACS_Rest_ = tACS applied for 40 s at motor rest, s-tACS_MSL_ = tACS applied for 40 s during motor activation, l-tACS_Rest_ = tACS applied for 20 min at motor rest, l-tACS_MSL_ = tACS applied for 20 min during motor activation. Mean and standard error of the mean are depicted.

**Figure 4 biomedicines-11-02218-f004:**
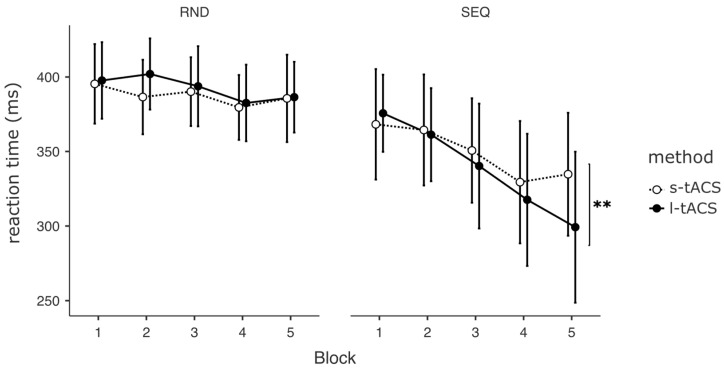
Reaction times of MSL during the experiment, shown for each block. RND = randomly presented stimuli, SEQ = sequentially presented stimuli, s-tACS = tACS applied for 40 s, l-tACS = tACS applied for 20 min. Mean and standard errors are depicted. ** *p* < 0.01.

**Figure 5 biomedicines-11-02218-f005:**
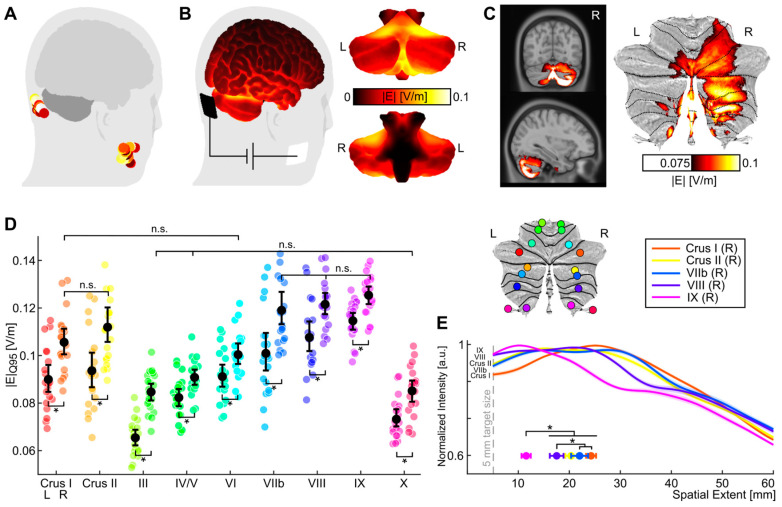
tACS montage and average electric field simulations. (**A**) Central points of the electrodes over the right cerebellum and over the right buccinator muscle were registered for each subject. (**B**) The average electric field magnitude was focused in posterior brain regions, especially in posterior areas of the right hemisphere of the cerebellum. (**C**) The whole-brain electric field showed the highest electric field intensities as depicted in an axial and sagittal slice (left). Descriptively, electric field magnitudes were highest in right areas IX, VIII, VIIb and Crus II and the respective vermis. (**D**) Analysis of electric field magnitudes (|E|_target_) in cerebellar ROIs confirmed significantly increased intensities in the right cerebellar hemisphere compared to the left cerebellar hemisphere (significant differences are indicated). The highest intensities were observed in right areas IX, VIII and VIIb, as intensities were analyzed across right ROIs (all pairwise comparisons were significant, except for the indicated non-significant pairs). Bootstrapped means and 95% confidence intervals are shown beside individual values. Median coordinates for each cerebellar ROI from the AAL atlas are depicted on a flatmap in the corresponding color. (**E**) For illustrative reasons, |E|_bias_ is depicted for those ROIs that show the highest intensities. The lowest bias was observed for area IX, followed by VIII. Overall results indicate robust targeting of cerebellar regions VIII and IX across subjects by the applied stimulation montage. Asterisks indicate *p* < 0.05 (Bonferroni-corrected), n.s., indicates no significance.

**Table 1 biomedicines-11-02218-t001:** Number of reported side effects with ratings > 0.

Side Effects	l-tACS_MSL_	l-tACS_Rest_	s-tACS_MSL_	s-tACS_Rest_	X^2^	*p*
Headache	4	2	4	6	2.00	0.572
Discomfort	5	4	5	5	0.16	0.984
Dizziness	0	1	1	1	0.00	1.000
Local heating	8	7	9	13	2.24	0.523

## Data Availability

The raw data supporting the conclusions of this article will be made available by the authors, without undue reservation.

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
