# Peer review of "Neuronavigated Cerebellar 50 Hz tACS: Attenuation of Stimulation Effects by Motor Sequence Learning"

_biomedicines, 2023, doi:10.3390/biomedicines11082218_

Round 1

Reviewer 1 Report

In this neurophysiologic study, Herzog et al. investigated the behavioural and neurophysiologic effects of short- and long-timed transcranial direct current stimulation (tACS) delivered over the right cerebellar hemisphere, in a cohort of young healthy subjects. Twenty participants were asked to perform four separate experimental sessions in which 40-second or 20-minute right cerebellar 50Hz tACS were delivered alone or during the execution of a modified serial reaction time task (i.e., the motor sequence learning - MSL). As a behavioural outcome, the authors included the reaction times required to complete an overall amount of 11 blocks (i.e., Simple - SMP, Random - RND and Sequence - SEQ). Conversely, as neurophysiologic measures, the authors assessed the amplitude of motor evoked potentials (MEPs), recorded from the right first dorsal interosseus muscle, before and at two time points after the experimental session. A post hoc assessment of the tACS-related electric field injected in the cerebella hemisphere was also performed. The results of the study overall pointed to an inhibitory effect of MSL on the tACS-induced plasticity effects, as shown by the decrease of MEP amplitudes following 20-minute tACS coupled with MSL. 

I believe that the study would help to gain the current knowledge on the tACS-related behavioural and neurophysiologic cerebellar function, under physiologic experimental conditions. I have also several concerns about the study designs, the results and the interpretation of findings. 

1. The study lacks experimental control conditions. For instance, the effects of MSL "per se" on MEP variability (i.e., a sham tACS condition) has not been considered. Also, the authors included only a single frequency of tACS (i.e., 50 Hz). It is known that the main relevant variable of tACS consists of frequency due to the entrainment of oscillatory neurons. This issue is relevant and the authors should discuss the pro-kinetic effects of 50Hz tACS (i.e., gamma tACS, not mentioned).  

2. The rationale for selecting 40-second and 20-minute tACS as short and long-timed stimulations, respectively, has not been clearly stated in the introduction. 

3. Crucial neurophysiologic parameters have not been reported and discussions, including the intensity of tACS stimulation, the resting motor threshold and active motor threshold, the intensity of the MSO for achieving 1mV MEP etc...

4. The study lacks significant clinical- behavioural- and neurophysiologic correlations. 

5. 15 MEPs have been recorded before and at two-time points following tACS: please specify the time points. Also, owing to the intrinsic MEP variability, 15 recordings would be few for assessing the corticospinal excitability. 

6. the reaction times of single fingers would significantly differ. How did the authors take into account that.?

7. Have been the MSL blocks correctly randomized in individual participants?

8. How did the authors consider the muscle fatigue potentially arising when performing 20-minute MSL, in the experimental paradigm?

Minor editing of English language required. 

Reviewer 2 Report

The current study had three purposes: 1) to determine the influence of cerebellar transcranial alternating current stimulation (c-tACS) on left primary motor cortex (M1) excitability and motor sequence learning (MSL); 2) to determine if MSL performed concurrent with c-tACS alters the effects of c-tACS; and 3) to evaluate neuronavigated electrode montages by post-hoc stimulation of individual electrical fields based on MRI data. A total of 20 healthy adults completed the study and participated in 4 experimental conditions a week apart in counterbalanced order. These conditions consisted of 50 Hz c-tACS (1 mA current peak to peak) of the right cerebellum applied for either 40 seconds (short tACS) or 20 minutes (long tACS) both at rest or while performing the MSL task. Motor evoked potentials were taken from the left M1 before stimulation as well as 5 and 49 minutes after stimulation. MSL consisted of a variation of the serial reaction time task of the right hand which has been used in numerous studies.

 The main findings were: 1) MEPS were increased after both short (40 sec) and long (20 min) c-tACS but only in the resting condition; and 2) MSL performance was increased only after the 20 minute c-tACS condition.

 Overall, I really liked this study as it was very comprehensive and well-done. It appeared to be conducted carefully, was easy to understand, and the methodology/design was strong (double blinded, use of neuronavigation, good TMS methodology, standard motor task, both rest and active state testing). I think the study adds greatly to the literature on the topics involved. The study should be of interest to readers of Biomedicines and researchers is several related fields.

 Thus, I think the study should be published. I only have a few comments, suggestions, clarifications, and changes the author should make before the paper is accepted. All of these are minor (see below).

 Introduction:

  1.  
    1. Line 62. Spell out 50 at the start of the sentence.
    2. Line 84. Space needed between “3cm”. There are other instances of similar typos in the paper, too many to point out individually, please proofread again.

 Methods:

  1.  
    1. It seems like in the second paragraph of the Methods that the authors should have mentioned that the short tACS condition was originally the sham condition instead of waiting until line 174 to say so. In addition, they should say why they did not call it that throughout the paper from the beginning. That could be done in the same paragraph or in the Discussion. I assume it is because they weren’t expecting a MEP increase in the sham condition? However, other tDCS studies have found small, but significant MEP increases with sham stimulation. Why not just say it early on instead of leaving the reader uncertain for much of the paper? I don’t think it affects the quality of the paper at all or the overall results if sham increased MEPs slightly.
    2. In the same paragraph, why not go ahead and let the reader know the time frames after stimulation that MEPS were taken? The reader had to wait and look way down the paper in a figure legend for that info.
    3. Line 171. It appears that the authors used conductive paste and did not have the rubber electrodes placed in saline soaked sponges. I feel that readers will want to know, why they chose that method, why they think it may or may not be better, and if they think that if sponges were used the results would be different. This could be added in a couple of sentences in the Discussion or in the associated paragraph. Since the author used a close variation of the Celnik montage, people who use this montage may be interested.
    4. Lines 177-178. Since this paragraph is only one sentence I recommend combining it into a previous paragraph.

 Results:

  1.  
    1. In Lines 278 and 280 as one example. Sometimes there is a zero before the decimal in statistical test reporting of the numbers and other times there is not. Probably should be consistent and always put the zero before the decimal.
    2. Figure 3. Consider putting asterisks indicating significance and where it occurred on the figure as was done for instance in Figure 4.

 Discussion:

  1.  
    1. Lines 362 to 364. Combine the one sentence paragraph with the next paragraph. Also the use of wording like “We could verify” should be improved. For instance, better grammar and use of tense would be “We verified” for instance. Also lines 408 and 409 are a one sentence paragraph.
    2. Lines 410 to 417. I believe a few studies by Nitsche and colleagues have shown that contractions done during tDCS obliterate the ability to see increases in excitability following tDCS of M1. The authors should consider mentioning this in this section as it would strengthen their results. For instance, https://pubmed.ncbi.nlm.nih.gov/21697590/
    3. Consider having a short limitation section near the end of the Discussion. For instance, the motor task could be a limitation as it is somewhat of a simple task and doesn’t have high accuracy requirements as discussed briefly in this paper https://pubmed.ncbi.nlm.nih.gov/21613597/. In other words, perhaps it could be mentioned that it is unknown if tACS can enhance more complicated tasks.
    4. Finally, it would probably be of great interest to the readers if the authors have any indication of how close to the Celnik montage stimulating electrode (3 cm lateral to inion) center was the average placement in the case of this study with the neuronavigation and considering the lobules the authors were trying to target. Especially since the other electrode was placed on the cheek as in the Celnik montage. For instance, did the center of the electrode in the current study end up being 2 cm lateral or 2 cm lateral and 1 cm up or whatever etc etc. If the authors have a way of knowing this information or measured it, I think it would be good to give a rough indication as many researchers who use the Celnik montage would likely be very interested in this information, especially for pilot studies were funding for MRI and neuronavigation may not be practical.

 There are a few errors in the bibliography where some titles of articles have all the words capitalized whereas others do not. For instance, compare references 20 vs 21 as one example. There are many other instances, ref 32 all caps the next 3 aren’t etc.

  1.  

 Most likely what happened is the authors used a program like Endnote and the program did not do what it was supposed to do 100% correctly with the bibliography. I have had it do it to me many times and have had to go back and fix some by hand. I assume it messed up here as well and the authors were not expecting the program to get basic formatting like that wrong since the difficult things it usually does well. Therefore, please proofread the bibliography for this issue.

some typos and grammatical errors but the amount is small.

Round 2

Reviewer 1 Report

I have no further concerns. 

Minor editing is required.